# Mechanical Properties of Fly Ash-Slag Based Geopolymer for Repair of Road Subgrade Diseases

**DOI:** 10.3390/polym15020309

**Published:** 2023-01-07

**Authors:** Jia Li, Xiaotian Dang, Jingwei Zhang, Peng Yi, Yongming Li

**Affiliations:** 1School of Water Conservancy and Civil Engineering, Zhengzhou University, Zhengzhou 450001, China; 2Henan Province Engineering Research Center for Infrastructure Repair and Reinforcement Materials, Zhengzhou 450000, China; 3JingweiBuilding Materials Corp., Ltd., Zhengzhou 450000, China; 4China Construction Eighth Engineering Division Corp., Ltd., Zhengzhou 450001, China

**Keywords:** geopolymer, road subgrade disease, solid activator, mechanical properties, microscopic structure

## Abstract

Fly ash-slag-based geopolymer is a grouting material with good fluidity and excellent mechanical and eco-friendly properties. The geopolymer can react chemically with the inert minerals of road subgrade under alkali excitation to form a good interfacial bond between road subgrade; therefore, it is suitable for the repair of weak road sections. In order to solve the problems such as the difficulty to store and transport the liquid activator of existing geopolymer grouting materials and to study the unclear mechanism of the influence factors on the fluidity and mechanical properties of geopolymer; the research on the mechanical properties of fly ash-slag based geopolymer was carried out in this paper. Experiments on the preparation of geopolymer and research on different ash-slag ratios under solid alkali excitation were studied. The influence of slag content and solid alkali content (NaOH, Na_2_SiO_3_) on the fluidity, compressive and flexural strength of fly ash-slag-based grouting materials was also researched on the basis of single-factor gradient tests. The results showed that the slurry fluidity decreased but the compressive strength gradually increased when the content of slag was increased from 20% to 50%. With the increase in alkali content (NaOH: 2–5%; sodium silicate: 0–6%), the slurry fluidity decreased and the compressive strength increased and then decreased. Combined with the analysis of the test results of Scanning Electron Microscopy (SEM), the microscopic structures of mechanical properties of geopolymer were investigated. Lastly based on ridge regression theory, a regression model was established to predict the mechanical properties of fly ash-slag-based geopolymer. The results indicate that fly ash-slag-based geopolymer has good mechanical properties and fluidity with proper contents of slag and alkali activator, which provide a reference for experiment research and engineering application.

## 1. Introduction

In China, road maintenance mileage accounts for 99% of the total road mileage. Every year, under the action of traffic loads and natural factors, highways have many diseases such as subgrade debonding (Figure 1a), subgrade weakness (Figure 1b) and road subsidence (Figure 1c). Aiming at road diseases, the commonly repaired grouting materials are mostly cement-based grouting materials, which have good durability and stability while having high carbon emissions and environmental pollution. In addition, cement-based grouting materials cannot react with the activation inert minerals of road subgrade, causing bad interface bonding and eventually subgrade diseases [1,2]. A relevant study shows that the production of cement requires the use of a large number of natural resources, and for every 1 ton of cement produced, 2.8 tons of raw materials are consumed [3]. During the production of cement, a large amount of CO_2_ is emitted into the atmosphere, and CO_2_ emissions from the cement industry have reached about 8% of global annual emissions [4]. Since the European Union released the European Green Deal in 2019 to take the lead in putting forward the goal of achieving carbon neutrality and related policies, countries such as the UK, Sweden, China, South Korea and the US have also put forward the goal of carbon neutrality one after another. In the big international process of carbon neutrality that has attracted global attention, there is an urgent need for grouting materials that can deal with road diseases and are green and low-carbon.

Geopolymer is a new type of gelling material with the main components of inorganic silica-oxygen tetrahedra and aluminum-oxygen octahedra. It has a three-dimensional network structure in space, which is formed by the polymerization of natural minerals, solid wastes and artificial silica-aluminum compounds as raw materials through strong alkali action and lattice reconstruction, as shown in Figure 2.

Due to its low density, low thermal conductivity, and low coefficient of temperature expansion, geopolymer has been widely used in aviation, civil engineering, ceramic arts, and nuclear waste containment engineering. With differences in raw materials and manufacturing processes, geopolymer has a wide type range and each has significant advantages and disadvantages, resulting in different suitable application fields. Research shows that fly ash, slag, construction waste and some other industrial wastes with low energy consumption can be used as raw materials for the development of geopolymer, which can save energy by more than 75% and reduce emissions by more than 95% [5,6]. So, the fly ash-slag-based geopolymer is the most widely used in civil engineering. The annual emission of slag in China’s iron and steel metallurgical industry is about 500 million tons, of which blast furnace slag accounts for as much as 50% [7]. Compared with the blast furnace slag resource utilization rate of 100% abroad (e.g., Japan, USA, Germany, etc.) [5], the Chinese resource utilization rate is slightly lower and most of the blast furnace slag is dumped, which is not only a waste of resources but also of great harm to the environment [8]. In fact, the micro-aggregate effect and volcanic ash effect of blast furnace slag milled into powder can optimize the matrix activity of cementitious materials and enhance the adhesion of mortar and aggregate by combining with an alkali activator [9]. Therefore it is a good choice to use slag as a precursor for geopolymer concrete.

As a common precursor of geopolymer material, the compressive strength of geopolymers formed from different types of fly ash is related to the variation of fly ash in particle size distribution, chemical composition, morphological properties and amorphous phase [10]. It has been shown that the geopolymer made from ultrafine fly ash has a higher compressive strength than conventional fly ash, but the flexural strength is only one-twentieth of that of conventional fly ash geopolymer [11]. In addition, calcium in fly ash can promote the formation of C-S-H gels, and fly ash with high calcium content is more conducive to improving the strength of geopolymer [12], but the inadequacy of calcium fraction on the development of three-dimensional mesh structure of geopolymer restricts the application of high-calcium fly ash [13]. Studies have shown that geopolymers with a high fly ash content tend to have better flowability but reduce the reaction rate and compressive strength [14]. Fly ash geopolymer can significantly improve the overall strength of the soil in the application of cured soil [15], but it has a slow reaction rate and low hardening strength, which make it difficult to achieve the bearing capacity requirements if it is directly applied in road subgrade repair. While slag is rich in calcium components, which have an extremely fast reaction rate under the effect of alkali activation, pure slag geopolymer has a very high hardening strength. Therefore, after alkali activation, the combination of fly ash and slag can not only ensure sufficient fluidity to fill the pores of the road base but also provide sufficient hardening strength to meet the bearing capacity requirements of roads. Studies have shown that the fly ash-slag geopolymer grouting materials can generate C-(A)-S-H, N-A-S-H and other gels to fill the pores of the subgrade to form a stable microstructure [16]. This can also form wraps around the subgrade soil particles and react with the inert minerals therein to a certain degree of polymerization to form a dense matrix and improve the interfacial bond [17], which is very beneficial to subgrade disease treatments.

For fly ash-slag-based geopolymer grouting materials, good mechanical properties such as flexural and compressive resistance are important to reinforce the subgrade and improve the overall bearing capacity of roads. However, a variety of factors affect the mechanical performance of the geopolymer, such as the modulus and concentration of alkali activator, Na_2_O content, slag content and other factors. Among them, the alkali activator modulus (the ratio of the material amounts of SiO_2_ and Na_2_O in the liquid alkali activator, which can be converted into NaOH content) and concentration are the most important factors affecting the mechanical strength of geopolymer nodules [18]. The variation in the alkali activator modulus can enhance the mechanical properties of the geopolymer, but higher or lower values than the optimal value will inhibit the strength rise instead [18]. The influence of slag content on mechanical properties is the second influence of the alkali activator modulus. With the increase in slag content, more C-S-H and C-A-S-H gels will be generated, which can significantly improve the compressive strength of fly ash geopolymer grouting materials [19], and the 28d strength of fly ash-slag geopolymer can reach 68 MPa with reasonable matching of all factors [20]. Some scholars have summarized the composition of geopolymer grouting materials and the way of enhancing the mechanical properties, which are in terms of raw materials, matching ratios and admixtures [21]. In order to further study the mechanism of influence of mechanical properties, many scholars have analyzed the microstructure of fly ash geopolymer grouting materials by means of XRD and SEM, showing that the incorporation of slag could improve the pore size distribution and generate a large number of sodium silica-aluminate and calcium silicate gels, which are conducive to the development of strength [22]. The appropriate amount of fly ash content is also beneficial to reduce the cracks generated by geopolymer hardening, making the structure more uniform, and improving the compressive strength of the geopolymer [23]. Some scholars have developed effective models that can predict the mechanical properties of geopolymers by Lasso regression and neural networks to clarify the variation pattern of the mechanical properties of geopolymer materials [24,25,26,27].

In addition, the fluidity of the geopolymer grouting materials determines whether the slurry can penetrate into the pores and cracks of the subgrade, therefore, determining the reinforcement effect of the subgrade. Many scholars’ studies have shown that the increase in slag and alkali activator content accelerates the polymerization reaction and generates a large amount of C-(A)-S-H gel in a short time and leads to poor flowability of the fly ash geopolymer slurry [28,29]. The synergistic effect of slag and alkali activator has also been considered, and the proportion of slag, alkali activator content and modulus has been investigated to ensure the mechanical properties of the fly ash-slag geopolymer grouting materials while enabling fluidity up to 23 s [30].

In the above-mentioned research, scholars mostly use the “Two-step Method- liquid activator” (Figure 3) to prepare geopolymers. Firstly, a liquid activator of the required modulus is prepared, then the raw silica-alumina material is reacted with the alkaline activator solution and the soil is cured. However, the “Two-step Method” geopolymer preparation requires an alkaline activator solution, which is difficult to transport and store, so the geopolymer cannot be applied in field construction. The “One-step Method-solid activator “ (Figure 3) geopolymer preparation mainly mixes silica-aluminum raw materials with a solid alkaline activator, and then adds distilled water to prepare a geopolymer slurry for curing clay [31]. This preparation process is suitable for the curing agent field construction technology, eliminating the environmental impact of the alkaline activator and reducing the transportation cost of the solution in field construction [31]. However, when the “One-step Method” has been used to prepare the slag-fly ash-based geopolymer as the curing agent, the raw material ratio, solid alkaline activator content and other factors have not been reported on the mechanical properties of the geopolymer at present.

In view of this, in order to solve the problem that the liquid exciter of existing geopolymer grouting materials is not easy to store, and the mechanism of the influence of solid exciter dosing and slag dosing on the fluidity and mechanical properties of geopolymer is not clear, solid alkali (NaOH, Na_2_SiO_3_) was used as the activator in this study to investigate the mechanical properties and fluidity of fly ash-slag based geopolymer grouting materials. The changes in the mechanical properties and fluidity with the influencing factors such as slag and solid alkali content were researched, and the microscopic morphology of specimens with different slag content and different alkali content with a Scanning Electron Microscope (SEM) were studied to explore the related mechanism. According to the test results, the ridge regression model of slag content, solid alkali content and mechanical properties of fly ash-slag-based geopolymer grouting materials was established to predict the macroscopic mechanical properties of geopolymer.

## 2. Materials and Methods

### 2.1. Experimental Materials

Fly ash-slag-based geopolymer is a class of inorganic cementitious materials with amorphous to quasi-crystalline structures formed by using alkaline activator (NaOH, Na_2_SiO_3_) and reactive silica-alumina materials (fly ash, slag, etc.) through geochemical-like reactions at room temperature. The raw materials in this paper are Class F fly ash (Figure 4a) with a specific surface area of 356.0 m^2^/kg (Table 1) and S95 grade slag (Figure 4b) with a specific surface area of 609.6 m^2^/kg (Table 2). The alkali activator used in the test is commercially available industrial grade flake NaOH (Figure 4e) with a 99.1% mass fraction (Table 3) and instant powder Na_2_SiO_3_ (Figure 4d), which contains 21.8% Na_2_O and 60.84% SiO_2_ (Table 4); the admixtures mainly include naphthalene water reducing agent which can improve the flowability of the slurry (Figure 4c) and a small amount of defoamer to improve the efficiency of defoaming and reduce the slurry bubbles.

### 2.2. Experimental Methods

#### 2.2.1. Geopolymer Preparation

Preparation: In this paper, the geopolymer is prepared by the “One-Step” method, in which a solid activator is used during the test, and the corresponding mass is weighed and mixed evenly with fly ash and slag according to the test design. After adding, we put the admixture into the mortar mixer and added water slowly to avoid slurry splashing. To ensure the complete dissolution of the solid activator, we mixed for 4 min to obtain the geopolymer grouting materials, poured them into a rectangular triplex mold of 40 mm × 40 mm × 160 mm without vibration and used a scraper to scrape the surface of the specimen, as shown in Figure 5.

In this paper, the content of two solid activators is converted into the concentration of Na_2_O and SiO_2_, as shown in the following equation:(1)$$C\left(\mathrm{NO}\right)=\frac{n_{\mathrm{NO}}}{v}$$
(2)$$n_{\mathrm{NO}}{=m}_{\mathrm{NH}}/{2M}_{\mathrm{NH}}{+m}_{\mathrm{NO}}/M_{\mathrm{NO}}$$
(3)$${v=m}_{w}/\rho_{w}$$
(4)$$C\left(\mathrm{SO}\right)=\frac{n_{\mathrm{SO}}}{v}=\frac{m_{\mathrm{SO}}/M_{\mathrm{SO}}}{m_{w}/\rho_{w}}$$
(5)$$n_{\mathrm{SO}}{=m}_{\mathrm{SO}}/M_{\mathrm{SO}}$$
(6)$${v=m}_{w}/\rho_{w}$$
where: C(NO), C(SO) are the concentrations of Na_2_O and SiO_2_ provided by the solid alkali activator; $n_{\mathrm{NO}}$, $n_{\mathrm{SO}}$ are the moles of Na_2_O and SiO_2_ provided by the solid alkali activator (mol); v is the volume of water (L); $m_{\mathrm{NH}}$, $M_{\mathrm{NH}}$ are the mass and molar mass of NaOH, respectively; $m_{\mathrm{NO}}$, $M_{\mathrm{NO}}$ are the mass and molar mass of Na_2_O in solid mass and molar mass of Na_2_O in Na_2_SiO_3_; $m_{\mathrm{SO}}$, $M_{\mathrm{SO}}$ are the mass and molar mass of SiO_2_ in solid Na_2_SiO_3_; $m_{w}$ the mass of the aqueous solution; $\rho_{w}$ the density of the aqueous solution.

Curing: ① After final solidification, we put the geopolymer into a standard curing box with a temperature of 20 ± 2 °C and a relative humidity ≥ 95%, and removed the mold after 1 day; ② specimens included standard ones and steaming ones, of which the standard specimens are put into the standard curing box until a specified time while the steaming ones are put in an 80 °C steaming box for 2 days.

#### 2.2.2. Test Program

Research [18,20,32] has shown that fly ash-slag-based geopolymer grouting materials with a 30%~50% slag and water binder ratio of 0.4~0.5 can guarantee mechanical properties and fluidity. So, in this paper the water binder ratio based on 40% slag content was pre-tested and it was found that the slurry fluidity exceeded 20 s when the water binder ratio is less than 0.42 (Figure 6a) and greater than 0.42 as there is a certain phenomenon of water secretion (Figure 6b). In order to ensure the fluidity of grouting materials and avoid the problem of water secretion, the water binder ratio is fixed at 0.42 in this paper.

In order to investigate the effect of the variation in the content of slag, NaOH and Na_2_SiO_3_ on the performance of fly ash-slag-based geopolymer grouting materials, the single-factor gradient test method is adopted in the paper. That is, each of the three test groups changes one variable and fixes the other two variables so as to investigate the influence law of this variable on the performance of geopolymer grouting materials.

In the test, the slag content is first changed and the contents of NaOH and Na_2_SiO_3_ are fixed. The slag content is set into four gradients from 20% to 50% to investigate its effect on the fluidity and mechanical properties and use it as an indicator to select the slag content. Secondly, the contents of slag and Na_2_SiO_3_ are fixed and the content of sodium hydroxide is varied from 2% to 5% to investigate its effect on the mechanical properties and flowability. Then the step is the same with Na_2_SiO_3_ and the specific experimental scheme is shown in Table 5.

#### 2.2.3. Fluidity Test

Referring to China Building Materials Industry Standard (JC/T0508-2005) “Cement Slurry Flow Determination Method” (flow cone method), the flow rate was defined as the time for the slurry flowing out of the flow cone. Firstly, the flow cone is placed vertically and the small opening below is blocked with a wooden plug, and the freshly prepared geopolymer grouting materials are poured into the flow cone with a total volume of 2000 mL to the calibration surface (Figure 7, net water flow rate of 8 s at the calibration surface). When the slurry flows out completely, the timing is stopped and the flow rate is recorded. Each group is measured three times to take the average value as the final flow rate.

#### 2.2.4. Mechanical Performance Test

Referring to the Chinese Standard (GB/17671-1999) “Cementitious Sand Strength Test Method”, the flexural and compressive properties of samples are tested by a cement mortar flexural/compressive tester with a flexural loading rate of 0.05 kN/s and a compressive loading rate of 2.4 kN/s (Figure 8, indication accuracy ± 1%, maximum flexural test force 10 kN, maximum compressive test force 300 kN). The abnormal values exceeding 15% of the average value are excluded, the average value of the remaining data is obtained, and the value is the representative value of the flexural and compressive strength of specimens.

#### 2.2.5. SEM Test

The specimen shall be made into sheet samples with a length and width less than or equal to 1 cm and thickness less than or equal to 1 cm. Then the samples are soaked in anhydrous ethanol for 48 h to stop hydration and the samples are removed and placed in a drying and curing box with a temperature of 40 °C after 48 h. After 24 h of drying, the samples are removed and sprayed with gold. Finally, the samples are placed in the sample chamber of the SEM- Zeiss Sigma 300 (Figure 9, resolution: 0.6 nm @ 15 kV, 1.0 nm @ 1 kV, magnification: 12×–2,000,000×, acceleration voltage: 0.02–30 kV) to analyze the microstructure of the geopolymer materials.

## 3. Results and Discussion

### 3.1. Compressive Strength

The proportion of raw materials and alkali activators are the most important factors affecting the mechanical properties of fly ash-slag-based geopolymer [19]. Therefore, three factors, namely, slag, NaOH and Na_2_SiO_3_, are considered to influence the flexural and compressive strength of grouting materials at different ages.

#### 3.1.1. Influence of Slag Content

The influence law of slag content on the compressive strength of the grouting materials is shown in Figure 10. According to the figure, it can be seen that the compressive strength of the geopolymer grouting materials increases with the increase in the curing age and the slag content. The analysis is as follows: First, the hydration degree of the geopolymer grouting materials becomes higher with curing age, so the compressive strength increases gradually with curing age. Second, while increasing the initial alkalinity of the solution, the slag also introduces calcium-rich substances into the fly ash system. Under the action of the alkali activator, the calcium-rich phase materials dissolve and release a large amount of Ca^2+^, which transforms the original Na_2_O-Al_2_O_3_-SiO_2_-H_2_O quaternary system into the Na_2_O-Al_2_O_3_-SiO_2_-CaO-H_2_O quintuple system. There are both two-dimensional chain structures (hydrated calcium silica/aluminate, hydrated calcium silica-aluminate) and three-dimensional mesh structures (hydrated sodium silica-aluminate, Figure 11) within the geopolymer system, which increase the disorder of the system structure and reduces the structural porosity, therefore, the compressive strength increases [33,34,35].

In addition, it can be found that the compressive strength under steaming conditions is higher than that under standard conditioning for 28d. This is because, in the high-temperature environment of steam curing, the materials have a higher activity to generate more reaction products and improve the degree of the polymerization reaction. However, the gap between the two strength values gradually decreases with the increase in slag content. This is because, for the fly ash-slag-based geopolymer grouting materials, the final strength is provided by fly ash and slag together, where low-activity fly ash can achieve a higher reaction level under the high-temperature environment of steam curing than that in the standard conditions. Moreover, with the increase in slag content, the fly ash in the system gradually decreases. At this time, compared with the standard curing, the fly ash that can be stimulated under steam curing is reduced, so the compressive strength gap between standard curing for 28d and steam curing is gradually narrowed.

Figure 12 shows the compressive damage of the geopolymer specimens with 30% and 50% slag content at the curing age of 3d and 28d. Through comparison, it can be found that the higher the content of slag, the longer the curing age, and the more serious the fragmentation when the geopolymer reaches the ultimate compressive strength. This also shows that the compressive strength of geopolymer has been increasing with the increase in slag content and the extension of curing age.

#### 3.1.2. Influence of Alkali Activator

Figure 13 shows the variation in the compressive strength of geopolymer grouting materials at different ages with different NaOH and Na_2_SiO_3_ contents. As can be seen from the figures, the effect of both on the compressive strength of geopolymer grouting materials is consistent, and the 28d and steaming strengths both show a trend of increasing first and then decreasing with the increase in NaOH and Na_2_SiO_3_ content. Additionally, the strength of steaming is higher than that of standardized 28d. From Figure 12a, it can be seen that the 28d compressive strength of the geopolymer slurry reaches the maximum value when the NaOH content is 3.5%, and the steam curing compressive strength reaches the maximum value when the NaOH content is 4.0%; from Figure 12b, it can be seen that the 28d compressive strength reaches the maximum value when the Na_2_SiO_3_ content is 3.0%, and the steam curing compressive strength reaches the maximum value when the Na_2_SiO_3_ content is 5.0%. However, the compressive strength of the geopolymer grouting materials at 1d, 3d and 7d does not change significantly with the increase in NaOH content but fluctuated with the increase in Na_2_SiO_3_ content. In addition, the compressive strength of the geopolymer grouting material increases with the increase in curing age when the content of NaOH and Na_2_SiO_3_ is constant. This is because the hydration degree of the geopolymer grouting material becomes higher and higher with the curing age, so the compressive strength gradually increases with the curing age.

In the process of alkali excitation, alkali solution dissolves fly ash particles and slag particles from the outside to the inside, and Ca^2+^, Si-O and Al-O tetrahedral monomers are successively dissolved from calcium-rich phase materials and silica-alumina-rich phase materials according to the activity level. The hydrated calcium silicate gel and hydrated calcium silicate/sodium aluminate gel are generated inside and outside the particles, which are filled inside the dissolved particles and around the still intact fly ash and slag particles. With the increase in gel products, they gradually wrap the particles and fill the gaps with each other, so that the overall structure gradually tends to be stable [36,37]. Therefore, for 28d and steaming strength, when the content of the alkali activator is low, the alkali environment provided by the alkali solution is weak. Moreover, the solubility of calcium-rich phase materials and silica-aluminum-rich phase materials is low, resulting in fewer reaction products to form a dense structure, so the strength is lower. With the increase in alkali activator content, the alkali environment is gradually enhanced, which increases the solubility of calcium-rich phase materials and silica-alumina-rich phase materials, resulting in more reaction products to improve the strength. However, when the content of the alkali activator is too high (NaOH content > 3.5%, Na_2_SiO_3_ content > 3.5%), the material mobility loss is too fast and the setting and hardening time is greatly advanced. That leads to a large number of fly ash particles and slag particles being wrapped by hydration products before dissolution, while the diffusion of Si-O and Al-O tetrahedral monomers that have been dissolved is inhibited and no further reaction can occur. So, the excessive alkali activator will reduce the degree of polymerization of the geopolymer and make its strength decrease. Therefore, with the increase in NaOH and Na_2_SiO_3_, the 28d and steaming strength show a trend of first increasing and then decreasing.

The calcium-rich phase in the slag determines that the early strength (1d, 3d, 7d) of the fly ash-slag-based geopolymer mainly comes from the slag. When the content of slag is certain (40%), NaOH changes from 2% to 5%, and the initial alkali environment within the slurry is also gradually enhanced, but the calcium phase materials provided by the slag are certain. So when the initial NaOH content can meet the reaction of the calcium phase materials, the early strength has reached a high level. Even if the NaOH continues to increase, the early strength is always maintained at the same level. The reason why the early strength fluctuates with the change in Na_2_SiO_3_ content is that when the content of Na_2_SiO_3_ < 3%, the C-S-H gel formed by the reaction between SiO_3_^2−^ and dissolved Ca^2+^ is also reduced, which is not conducive to the growth of early strength. When the Na_2_SiO_3_ content > 3%, the surplus SiO_3_^2−^ forms mSiO_2_-nH_2_O (silicate gel) attached to the surface of mineral powder and fly ash particles, which will hinder the dissolution of Ca^2+^ and reduce the amount of C-S-H gel generation [38,39]. Therefore, when the Na_2_SiO_3_ content is varied from 1% to 6%, the early strength of the geopolymer grouting materials has a fluctuating change that first slightly increases and then decreases.

In addition, it can be seen from Figure 13 that the difference between the compressive strength of steam curing and standard 28d curing gradually increases with the increase in alkali activator content. This is because the precursor materials have a higher activity to generate more reaction products under the high-temperature environment provided by the steaming, which increases the degree of polymerization, then forms a higher strength [40]. With the increase in the content of the alkali activator, the alkali in the geopolymer system is gradually sufficient and steaming materials can reach a higher level of reaction and polymerization. So, there is a large gap between the compressive strength of steaming and the 28d standard condition. However, when the content of precursor materials is certain if the content of the alkali activator is too low to completely excite the activity of precursor materials, even in the high steaming temperature environment, the materials cannot reach a higher level of reaction and polymerization, so the compressive strength of the steaming and 28d specimen is closer. 

Figure 14 shows the compressive damage of geopolymer specimens with different alkali activator content at the 28d curing age. Through comparison, it can be found that when the ultimate compressive strength is destroyed, the geopolymer specimens with more serious fragmentation tend to have a higher compressive strength, which is consistent with the upward trend of 28d compressive strength of geopolymer when NaOH is from 2% to 3.5%, and Na_2_SiO_3_ is from 0% to 3%.

### 3.2. Flexural Strength

#### 3.2.1. Influence of Slag Content

The effect of slag content on the flexural strength of fly ash-slag-based geopolymer grouting materials at different ages is shown in Figure 15. It can be seen from the diagram that when the slag content is 20%, the flexural strength gradually increases with the curing age, which is because the content of slag is not high at this time, and the resulting C-S-H gel is not excessive, which can improve the flexural strength without destroying the structure of the geopolymer. When the slag content is between 30% and 50%, the flexural strength first increases and then decreases with the curing age, reaching the maximum value on the seventh day. This is because a large number of C-S-H gels are generated when the slag is mixed. The C-S-H gel is beneficial to the flexural strength in the early stage of maintenance, but the degree of slag hydration increases with the prolongation of curing age, and the excessive C-S-H gel hardening increases the brittleness of the geopolymer specimen while destroying the three-dimensional mesh structure of the geopolymer, leading to micro-cracks in the geopolymer structure and the decreases in flexural strength. For the specimens of the same age, the flexural strength at 1d, 3d and 7d is increasing with the increase in slag content, while the flexural strength at 28d and steaming increases first and then decreases. This is because the early flexural strength (less than or equal to 7d) of fly ash-slag-based geopolymer grouting materials mainly originates from the C-S-H gel in slag, so the increase in slag content can improve the early flexural strength. However, the longer the curing time, the greater the brittleness development in the geopolymer with a large slag content, so the flexural strength of 28d and steaming show the law of first increasing and then decreasing.

#### 3.2.2. Influence of Alkali Activator Content

Figure 16, respectively, shows the variation in the flexural strength of the fly ash-slag-based geopolymer grouting materials at different ages with different NaOH and Na_2_SiO_3_ content. When the content of the alkali activator is low (NaOH ≤ 3%, Na_2_SiO_3_ ≤ 2%), the flexural strength increases and then decreases with the increases in curing age, and all of them reach the maximum value on the seventh curing day. When the content of alkali is high (3.5% ≤ NaOH ≤ 5%, 3% ≤ Na_2_SiO_3_ ≤ 6%), flexural strength tends to increase with the curing age. This may be due to the low content of the alkali activator, and therefore, the low reaction rate of the geopolymer and the sufficient time to form a dense spatial structure during the hydration of C-S-H gels and sodium silicate gels [41]. Then the early flexural strength increases significantly. However, with the curing age, the C-S-H gels generated by subsequent hydration destroy the original dense spatial structure, resulting in a decrease in flexural strength. When the content of alkali exciter is high, the reaction rate of geopolymer is faster, then the C-S-H gel and sodium silicate gel are hardened before the dense spatial structure is fully formed. Moreover, the degree of the polymerization reaction is reduced, plus specimens continue to produce hydration products in the process of maintenance, so the flexural strength as a whole shows a rising trend with the maintenance age, and does not appear in the larger values in the early maintenance.

### 3.3. Fluidity

The flow rate of the slurry is defined as the time required for the slurry to flow out of the inverted cone in this paper. For grouting materials, the flow rate requirement (less than or equal to 20 s) has to be met to achieve a better filling bonding effect.

#### 3.3.1. Influence of Slag Content

Figure 17 shows the flow rate of the geopolymer grouting materials with different slag content. It can be seen from the figure: when the alkali content and water-cement ratio are constant and the slag content grows from 20% to 50%, the flow rate of the slurry increases and the liquidity of the slurry gradually decreases.

This is due to the physical water absorption characteristics of slag, increasing the slurry consistency and decreasing the fly ash content, then weakening the “micro-bead effect”. The so-called “micro-bead effect” is that fly ash contains a large number of glass beads of which the particle form is mainly spherical, as shown in Figure 18. After mixing, the “micro-bead” can be uniformly distributed in the slurry system and increase the fluidity of the slurry, so the increase in the content of slag weakens the “micro-bead effect” of fly ash, resulting in the decrease in slurry fluidity. From the analysis of the reaction mechanism, the addition of slag increases the content of calcium-rich phase material in the system, also increasing the initial alkalinity of the slurry [42]. Under the alkaline environment, the calcium-rich phase has lower bond energy, higher activity and faster dissolution, so the calcium-rich phase can quickly react violently with OH^−^ and SiO_3_^2−^ in the solution to produce Ca(OH)_2_ precipitate and hydrated calcium silicate gel [43,44], thus increasing the consistency of the slurry.

Figure 19 shows the flow state of free diffused slurry with different slag content at 30 s. From the previous analysis, it can be seen that with the increase in slag content, the consistency of the geopolymer grouting materials increases while the flowability decreases. The slag content increases from 20% to 50% while the diffusion range of slurry decreases from 220 mm to 150 mm.

#### 3.3.2. Influence of Alkali Activator

Figure 20 shows the effects of different content of NaOH and Na_2_SiO_3_ on the flowability of the geopolymer slurry. From the figure, it can be seen that the flow rate of slurry tends to increase with the increase in NaOH and Na_2_SiO_3_ content, and the slurry fluidity gradually decreases.

Davidovits put forward that the reaction process of geopolymer materials in the presence of an alkali activator includes four steps: dissolution, diffusion, condensation, and hardening, and the increase in alkali activator increases the rate of the polymerization reaction [45]. The higher the alkali content, the higher the initial OH^−^ and SiO_3_^2−^ concentrations, and the faster the condensation and hardening process of the geopolymer materials, therefore, the fluidity decreases. The reason can be explained as follows: the increasing solubility of the silicon-aluminum phase and calcium-rich phase components in the fly ash and slag lead to released Ca^2+^, Si-O and Al-O tetrahedral monomers diffusing into the solution and forming hydrated aluminate and hydrated silicate monomers through chemical hydration reactions and physical electrostatic reactions [46]. At the same time, some free water is converted into bound water, which accelerates the polycondensation and hardening process of the geopolymer materials. Therefore, an increase in the content of alkali accelerates the reaction process of the geopolymer, generates more hydration products and reduces the free water content, resulting in an increase in the time for the complete flow of the slurry and a decrease in the fluidity of the geopolymer grouting materials.

Figure 21 shows the flow state of the slurry with different alkali activator content at 30 s. From (a) and (b), the diffusion range of slurry decreases from 245 mm to 160 mm with NaOH contents changing from 2% to 5%. Moreover, from (c) and (d), the diffusion range of slurry decreases from 210 mm to 170 mm with the Na_2_SiO_3_ content changing from 0 to 6%. This also shows that the initial state of the slurry gradually deteriorates and the flowability of the slurry gradually decreases with the increase in the alkali activator content.

### 3.4. Microscopic Morphological Analysis

In this paper, SEM is used to analyze the microscopic morphology of 28d geopolymer grouting materials with different slag and alkali activator contents to study the morphology of the hydration products of geopolymer slurry hardening, which reveal the microscopic formation of its mechanical properties.

#### 3.4.1. Influence of Slag Content

Figure 22 illustrates the microscopic morphology of the geopolymer slurry of 28d with different slag content. The microscopic morphology at 20%, 30%, 40% and 50% of the slag content can be seen in the figure. With the increase in slag content, the observable “holes” and uncoated fly ash particles are less and less, and the structure of the geopolymer slurry is gradually dense. When the content of slag is 20%, a large number of spherical fly ash particles that are not dissolved and not wrapped by hydration products can be observed. This is due to the low activity of fly ash and slag. The hydration products are mainly sodium silicate (N-A-S-H), even if there is a sufficient amount of alkali which cannot produce enough N-A-S-H to fill the structural pores, resulting in the overall loose structure, so the mechanical properties are poor.

When the slag content gradually increases to 50%, the active materials in the geopolymer system increase to provide a large amount of Ca^2+^, and the gel products such as hydrated calcium silicate (C-S-H), hydrated calcium aluminate (C-A-H) and hydrated calcium silicate aluminate (C-(A)-S-H) are gradually increased. Then, the reaction level and polymerization degree of the system is gradually improved. The internal hydration products of the geopolymer are sufficient to fill the pores between the undissolved particles. Moreover, the structure of the geopolymer slurry becomes denser, so the mechanical properties are gradually improved with the increase in slag content. Therefore, the increase in slag content can increase the gels of hydrated calcium silicate (C-S-H), hydrated calcium aluminate (C-A-H) and hydrated calcium silicate aluminate (C-(A)-S-H) in the system, increase the disorder in the system structure and reduce the structural porosity. Then the compressive strength of the geopolymer grouting materials is improved, which is consistent with the law obtained in Figure 9.

#### 3.4.2. Influence of Alkali Activator Content

Figure 23 and Figure 24 show the microscopic morphology of the geopolymer slurry of 28d with different NaOH and Na_2_SiO_3_ content. As can be seen from Figure 23, the microscopic morphology of the NaOH content at 2%, 3.5%, and 5% can be seen. With the increase in NaOH content, the observable “holes” decrease first and then increase, and the structure of the geopolymer slurry is dense first and then loose. Although there is a sufficient amount of slag and Na_2_SiO_3_ in the system, the overall alkali environment is weak when the NaOH content is 2%, which is not enough to fully excite the active substances within raw materials. At this time, the reaction degree is low, and the gel products such as hydrated sodium silicate aluminate (N-A-S-H) and hydrated calcium silicate (C-S-H) are insufficient, resulting in more voids inside. When the NaOH content is 3.5% and 5%, the gel products such as calcium silicate hydrate (C-S-H) and calcium aluminosilicate hydrate (C-(A)-S-H) increase, and the denseness of the structure is significantly improved. However, compared with the former, there are still some “holes” and “pits” in the microscopic morphology of the latter. This is most likely because the overall reaction process is accelerated by the excessive amount of alkali used. The dissolved active material and the generated hydration products have been condensed and hardened before they spread evenly, and the unresolved particles wrapped in them cannot continue to be dissolved, so the overall reaction degree is reduced, resulting in these “holes” and “pits”.

As can be seen from Figure 24, when the Na_2_SiO_3_ content is from 0 to 3%, the microstructure of the geopolymer slurry is dense, but its content is high which also reduces the degree of polymerization, resulting in a large number of “holes” that are not conducive to the structure. In addition, from Figure 23 and Figure 24 we find that the variation in NaOH content (e.g., from 2% to 3.5%) has a greater effect on the structural denseness of the geopolymer slurry than sodium silicate (e.g., from 0 to 3%). This may be due to the fact that NaOH is more alkaline and directly affects the formation of hydration products.

From the microscopic results, it can be seen that when the content of the alkali activator is low, the reaction products in the system are few and cannot form a dense structure, so the strength of the geopolymer materials is low. With the increase in alkali activator content, the reaction products gradually increase, which in turn, improves the strength of the materials. When the content of the alkali activator is high, the speed of solidification and hardening of geopolymer is accelerated, which inhibits the hydration reaction of slag and fly ash, resulting in a lower degree of polymerization of geopolymer, so the strength of the material decreases. Therefore, the mechanical properties of the geopolymer slurry increase and then decrease with the increase in NaOH and Na_2_SiO_3_ content. The microscopic performance of NaOH and Na_2_SiO_3_ is consistent with the macroscopic change law of compressive strength in Figure 12.

## 4. Model of Compressive Strength

In this section, the model of the unconfined compressive strength of fly ash-slag-based geopolymer grouting materials is constructed based on the ridge regression theory, and the validity of the model is verified based on the test results.

### 4.1. Ridge Regression Modeling

In this paper, a ridge regression algorithm with L2 regularization is used to establish a regression prediction model for the unconfined compressive strength of fly ash-slag-based geopolymer grouting materials at different ages. Among them, the multiple regression equations for modeling are as follows:(7)$${y=a}_{1}x_{1}{+a}_{2}x_{2\;}^{2}{+\ldots +a}_{n}x_{n\;}^{n}+b$$
where: y is the target value; $x_{1}$, $x_{2\;}$... $x_{n}$ are the eigenvalues; $a_{1}$, $a_{2\;}$... $a_{n}$ are the coefficients and b is the constant term. 

The coefficients of the multiple regression equation in the ridge regression algorithm are solved using a loss function with L2 regularization (see Equation (2)), which can weaken the effect of higher-order features while preserving all model features. The coefficients are solved when the sum of squared errors between the predicted and actual values is minimized.
(8)$$J_{L2}=\sum_{i=1}^{m}{\left(y\left(x_{i}\right)-y_{i}\right)}^{2}+\lambda \sum_{i=1}^{m}a_{i}^{2}$$
where: $J_{L2}$ is the total loss; $y\left(x_{i}\right)$ is the predicted value of the ith training sample; $y_{i}$ is the true value of the ith training sample; $\sum_{i=1}^{m}\left|a_{i}\right|,\;\sum_{i=1}^{m}a_{i}^{2}$ are the penalty terms; λ is the penalty coefficient.

In this paper, the slag content, NaOH content and Na_2_SiO_3_ content are taken as the basic characteristic parameters and the unconfined compressive strength of each age is taken as the target value Y. To ensure the accuracy and generalization of the model, the basic characteristics are quadratic and standardized. The weight coefficients of each feature obtained and bias coefficients from the final model are shown in Table 6.

### 4.2. Model Validation

In this paper, the coefficient of determination (COD) and mean square error (MSE) are used as indicators to evaluate the prediction accuracy of the ridge regression model. The evaluation indexes of the prediction models for each age are shown in Table 7 and Figure 25.

From Figure 25a,d, it can be seen that the model has better prediction results for 1d and 28d strength in the training set and the COD values of the model are 0.82 and 0.88, respectively. From Figure 25b,c, the model has a slight error in prediction results for 3d and 7d strength in the training set, and the COD values of the model are 0.76 and 0.74, respectively. Meanwhile, it can be seen from Figure 25 that the model has better prediction results for 1d, 3d, 7d and 28d strength in the test set, and the COD values are 0.88, 0.89, 0.86 and 0.9, respectively. Comparing the results of the training set with those of the test set, it can be found that the model has lower MSE and COD values greater than 0.85 in the test set for each age, indicating that the prediction models have better prediction results in the test set and has good generalization ability.

Figure 26 shows the predicted value and actual value change of the unconfined compressive strength of the fly ash-slag-based geopolymer grouting materials of each age by the ridge regression prediction model. As can be seen from the figure, the predicted curves are very close to the measured curves, with COD values of 0.88, 0.89, 0.86, and 0.91 for each age (1d, 3d, 7d, and 28d) and MSE values of 0.14, 0.23, 0.44, and 0.73, respectively. The higher COD values and smaller MES values result in a good agreement between the prediction curves and the measured curves in terms of variation laws and values, which indicates that the model can achieve the prediction of the unconfined compressive strength of fly ash-slag-based geopolymer grouting materials at each age.

## 5. Conclusions

In this paper, the mechanical properties and flowability of fly ash-slag-based geopolymer grouting materials are studied. Combined with the experimental data, a mechanical property prediction model is established based on the ridge regression theory to realize the prediction of the unconfined compressive strength of fly ash-slag-based geopolymer grouting materials at each age. The conclusions are as follows:(1)The influence law and mechanism of the unconfined compressive strength of geopolymer grouting materials are following:

The increase in slag content can generate a large number of gels such as C-S-H and C-A-S-H to increase the disorder of the geopolymer system, thus improving the unconfined compressive strength of the geopolymer grouting material at all ages. With the increase in alkali activator (NaOH, Na_2_SiO_3_), the degree of polymerization increases first and then decreases, so the 28d unconfined compressive strength increases first and then decreases; the compressive strength of the geopolymer grouting materials at 1d, 3d and 7d did not change significantly with the increase in NaOH content but fluctuated with the increase in Na_2_SiO_3_ content. The microscopic structures based on SEM results verify the influence mechanism of mechanical properties. A regression prediction model of the unconfined compressive strength of different ages is established in the paper, which has good agreement with experimental data.

(2)The influence law and mechanism of the flexural strength of geopolymer grouting material are as follows:

With the increase in slag, the flexural strength of the geopolymer of 1d, 3d and 7d increases and reaches the maximum value on the seventh day. The flexural strength of 28d and steaming first increases and then decreases. The microscopic structures based on SEM results verify the influence mechanism of flexural strength. When the content of the alkali activator is low (NaOH ≤ 3%, Na_2_SiO_3_ ≤ 2%), the flexural strength increases and then decreases with the increases in curing age, and all of them reach the maximum value on the seventh curing day. When the content of alkali is high (3.5% ≤ NaOH ≤ 5%, 3% ≤ Na_2_SiO_3_ ≤ 6%), flexural strength tends to increase with the curing age.

(3)The influence law and mechanism of the fluidity of geopolymer grouting materials are following:

The fluidity of the slurry gradually decreases with the increase in slag content. This is because the increase in slag content not only weakens the “micro-bead effect” of fly ash but also introduces a large amount of calcium-rich phase materials, which promotes the generation of Ca(OH)_2_ precipitation and hydrated calcium silicate gel, increasing the consistency of the slurry. With the increase in alkali activator content, the fluidity of the slurry also gradually decreases. This is because the increase in alkali activator accelerates the reaction process of the geopolymer, which generates more hydration products at the same time and reduces the free water content, leading to the decrease in slurry fluidity.

## Figures and Tables

**Figure 1 polymers-15-00309-f001:**
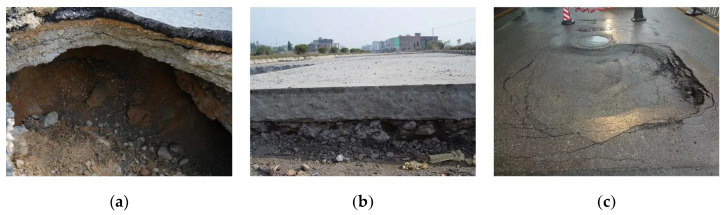
Common road diseases. (**a**) Subgrade debonding; (**b**) Subgrade weakness; (**c**) Road subsidence.

**Figure 2 polymers-15-00309-f002:**
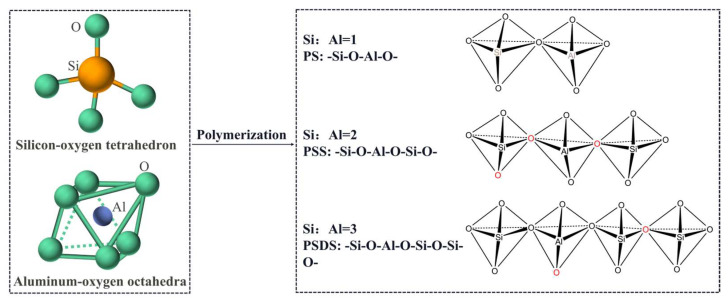
Schematic diagram of geopolymer structure.

**Figure 3 polymers-15-00309-f003:**
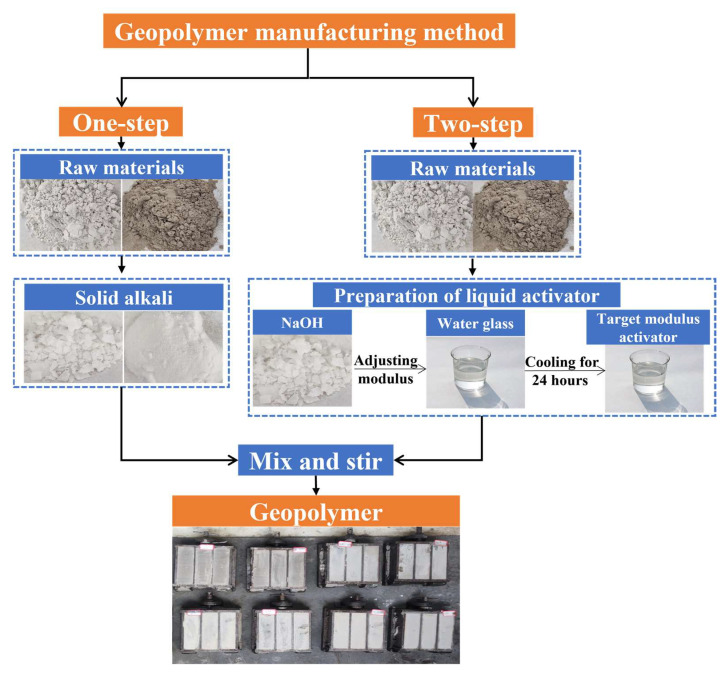
Preparation method of geopolymer.

**Figure 4 polymers-15-00309-f004:**
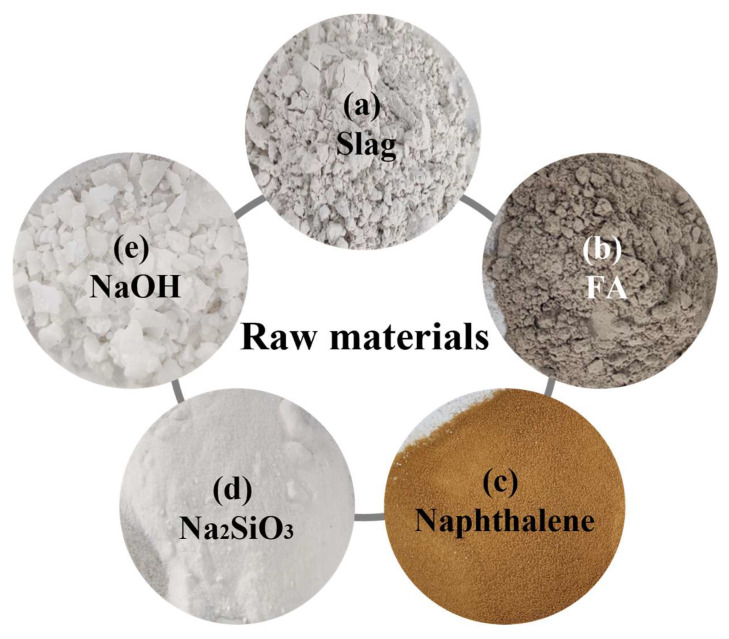
Test materials.

**Figure 5 polymers-15-00309-f005:**
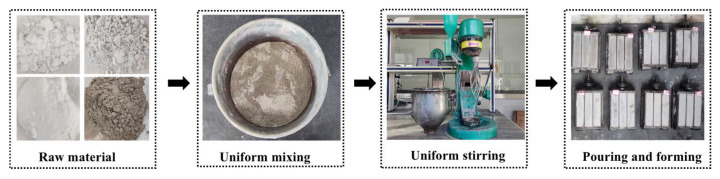
Specimen preparation of the geopolymer grouting materials.

**Figure 6 polymers-15-00309-f006:**
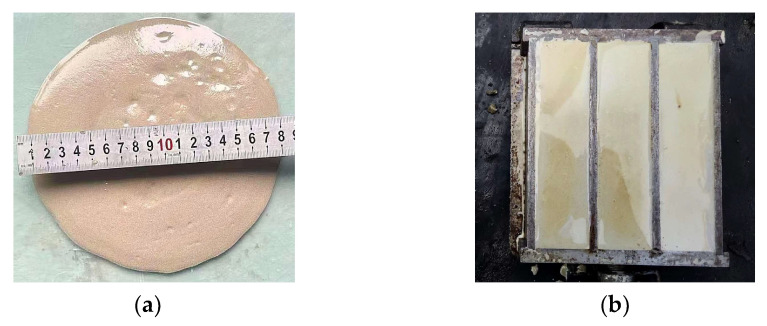
Pre-test of water binder ratio. (**a**) Fluidity; (**b**) Water secretion.

**Figure 7 polymers-15-00309-f007:**
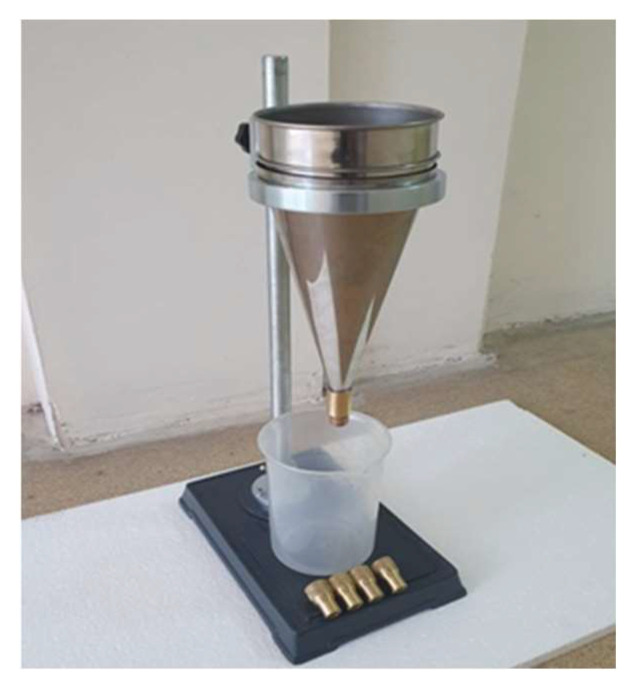
Flow cone.

**Figure 8 polymers-15-00309-f008:**
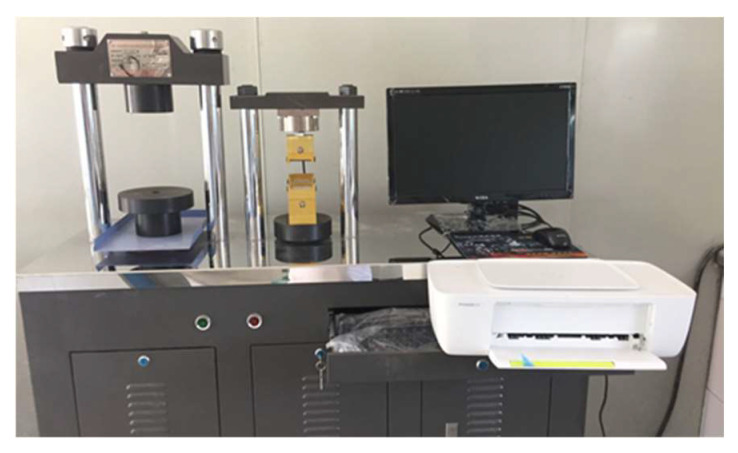
Cement mortar flexural/compressive tester.

**Figure 9 polymers-15-00309-f009:**
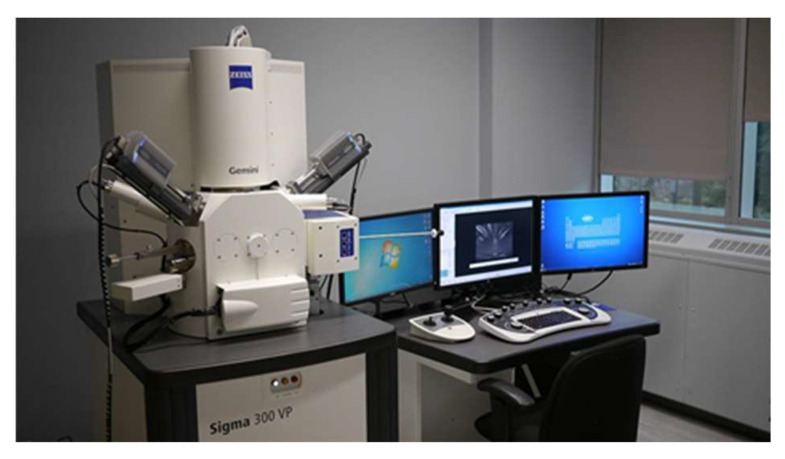
Zeiss Sigma 300.

**Figure 10 polymers-15-00309-f010:**
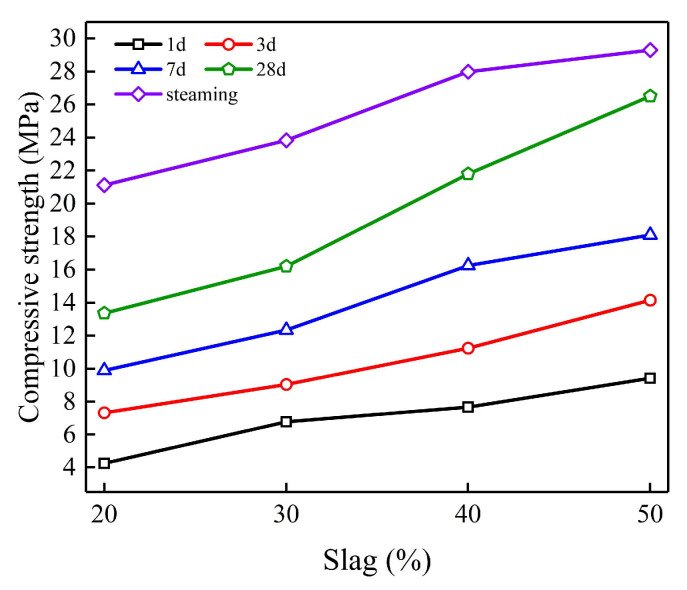
Effect of slag content on compressive strength.

**Figure 11 polymers-15-00309-f011:**
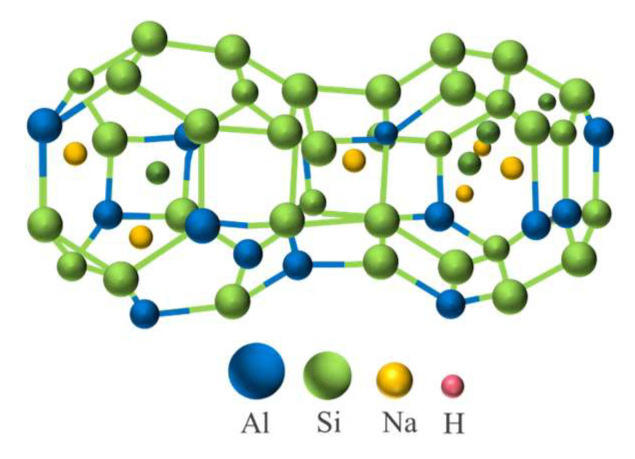
Three-dimensional reticulated silica-aluminate.

**Figure 12 polymers-15-00309-f012:**
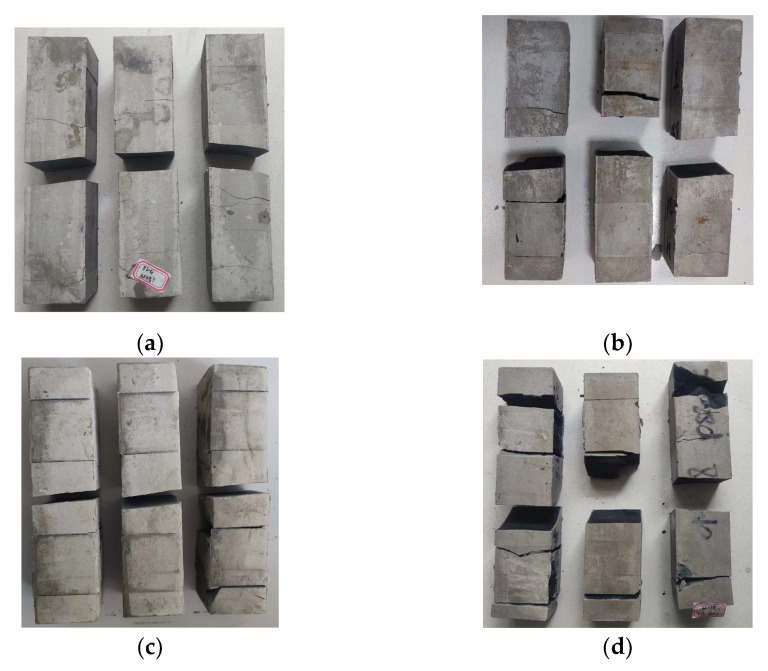
Compressive damage of geopolymer specimens with different slag content and age. (**a**) Slag content: 30%, age: 3d; (**b**) Slag content: 30%, age: 28d; (**c**) Slag content: 50%, age: 3d; (**d**) Slag content: 50%, age: 28d.

**Figure 13 polymers-15-00309-f013:**
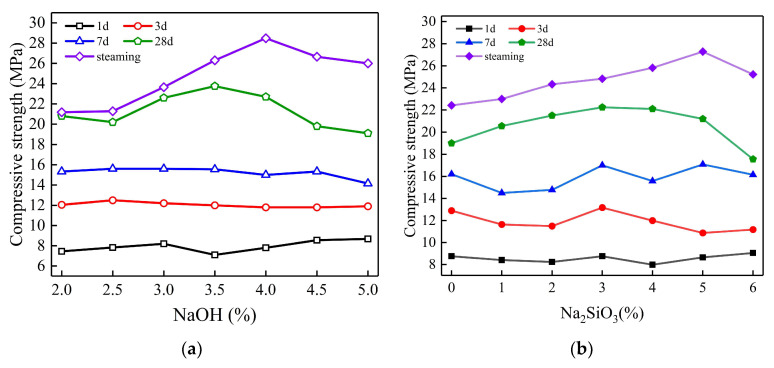
Effect of alkali activator on compressive strength. (**a**) Effect of NaOH content; (**b**) Effect of Na_2_SiO_3_ content.

**Figure 14 polymers-15-00309-f014:**
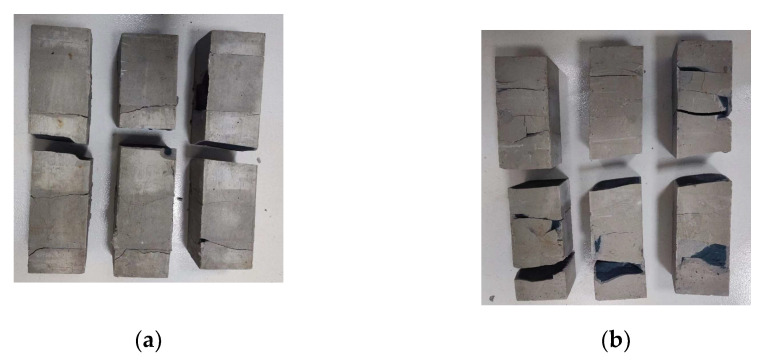
Twenty-eight day compressive damage of geopolymer specimens with different alkali activator content. Content of alkali activator %: (**a**) NaOH: 2%; (**b**) NaOH: 3.5%; (**c**) Na_2_SiO_3_: 0%; (**d**) Na_2_SiO_3_: 3%.

**Figure 15 polymers-15-00309-f015:**
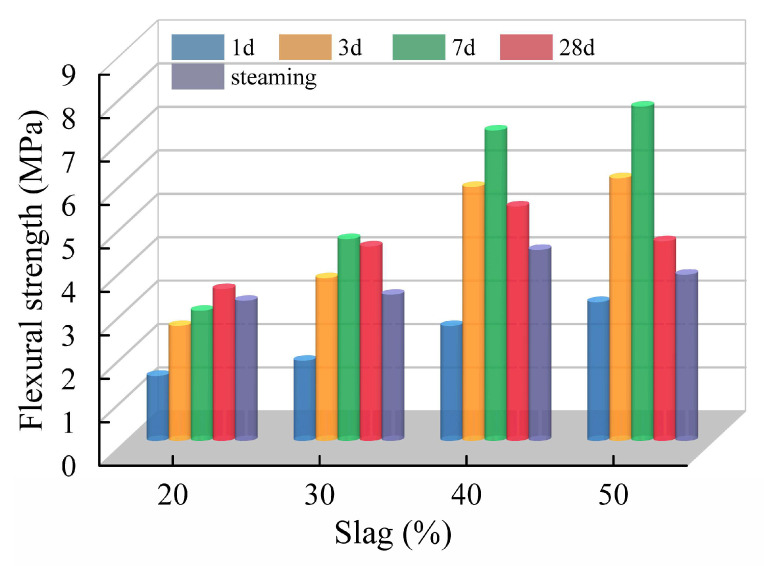
Effect of slag content on flexural strength.

**Figure 16 polymers-15-00309-f016:**
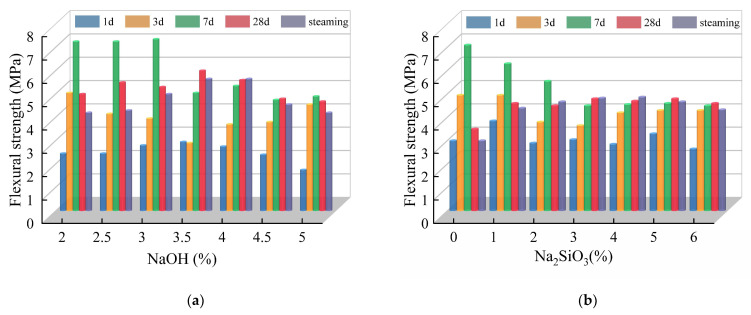
Effect of alkali activator content on flexural strength: (**a**) NaOH; (**b**) Na_2_SiO_3_.

**Figure 17 polymers-15-00309-f017:**
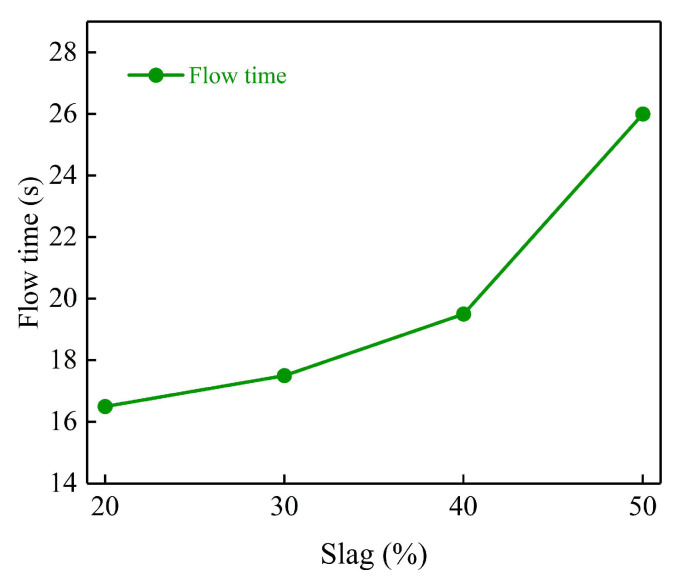
Effect of different mineral powder blending on flow rate.

**Figure 18 polymers-15-00309-f018:**
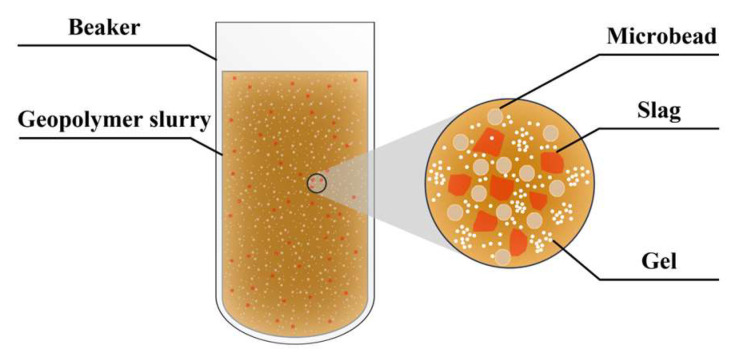
Micro-bead effect.

**Figure 19 polymers-15-00309-f019:**
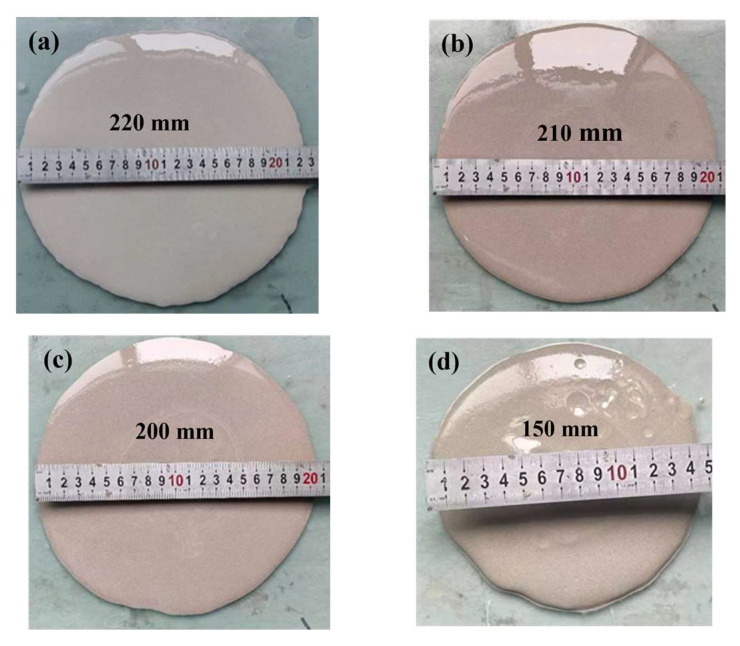
Flow state of slurry at 30 s with different slag content: (**a**) 20%; (**b**) 30%; (**c**) 40%; (**d**) 50%.

**Figure 20 polymers-15-00309-f020:**
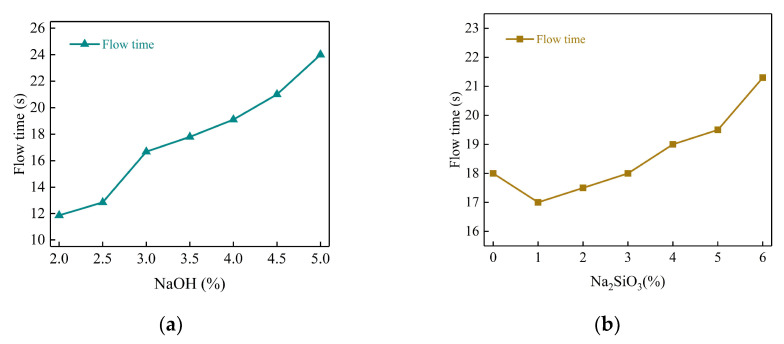
Effect of alkali activator on fluidity. (**a**) NaOH; (**b**) Na_2_SiO_3_.

**Figure 21 polymers-15-00309-f021:**
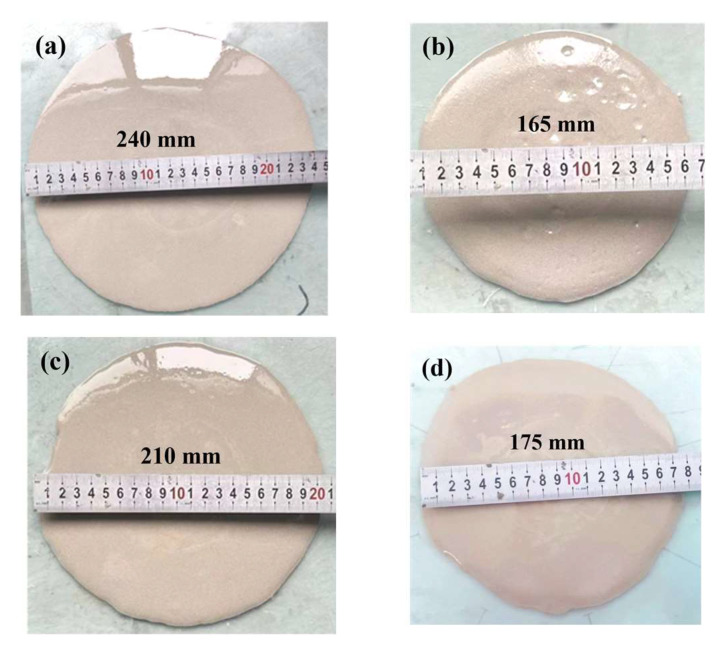
Flow state of slurry at 30 s with different alkali activator content: (**a**) NaOH: 2%; (**b**) NaOH: 5%; (**c**) Na_2_SiO_3_: 0%; (**d**) Na_2_SiO_3_: 6%.

**Figure 22 polymers-15-00309-f022:**
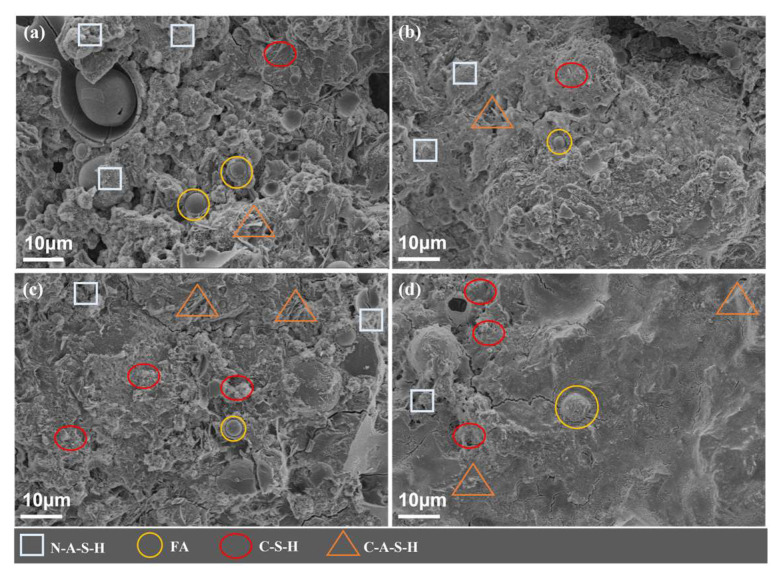
SEM morphology of the geopolymer slurry with different slag content (28d, NaOH: 4%, Na_2_SiO_3_: 5%). (**a**) Slag content: 20%; (**b**) Slag content: 30%; (**c**) Slag content: 40%; (**d**) Slag content: 50%.

**Figure 23 polymers-15-00309-f023:**
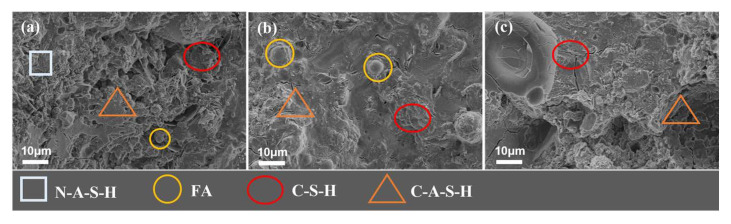
SEM morphology of geopolymer slurry with different NaOH content (28d, slag: 40%, Na_2_SiO_3_: 5%). (**a**) NaOH content: 2%; (**b**) NaOH content: 3.5%; (**c**) NaOH content: 5%.

**Figure 24 polymers-15-00309-f024:**
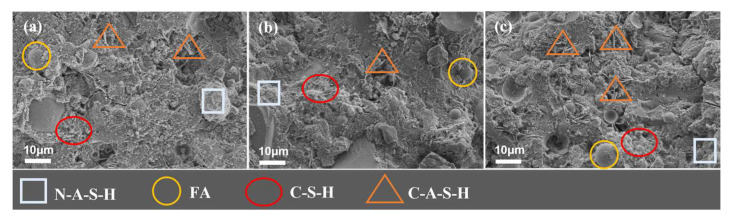
SEM morphology of geopolymer slurry with different Na_2_SiO_3_ content (28d, slag: 40%, NaOH: 4%). (**a**) Na_2_SiO_3_ content: 0%; (**b**) Na_2_SiO_3_ content: 3%; (**c**) Na_2_SiO_3_ content: 6%.

**Figure 25 polymers-15-00309-f025:**
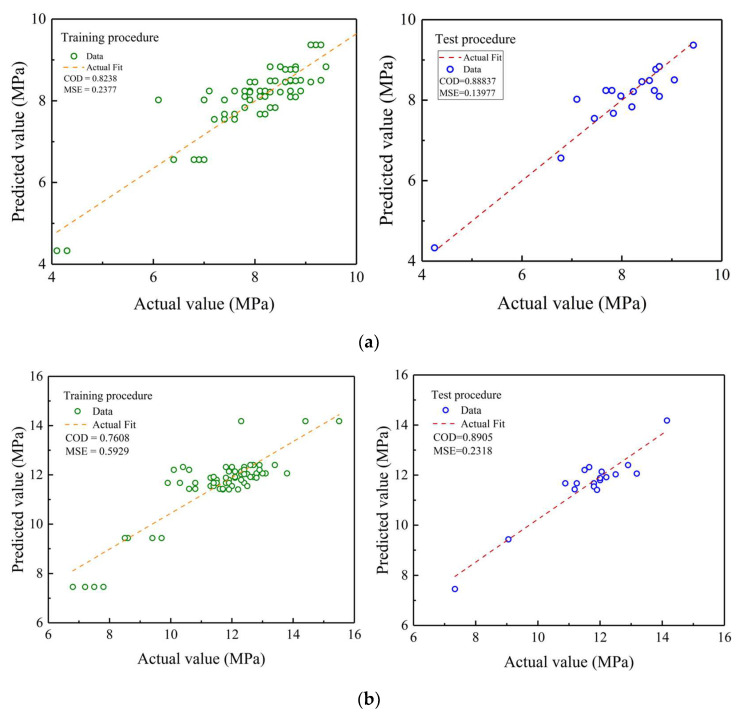
Predicted and actual value of compressive strength of each age. (**a**) Predicted value and actual value of compressive strength of 1d; (**b**) Predicted value and actual value compressive strength of 3d; (**c**) Predicted value and actual value of compressive strength of 7d; (**d**) Predicted value and actual value of compressive strength of 28d.

**Figure 26 polymers-15-00309-f026:**
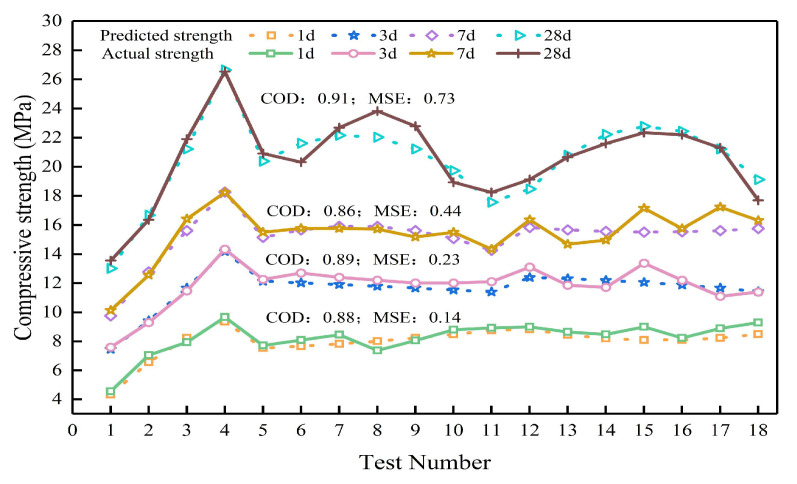
Variation curves of predicted and measured value of models with different age.

**Table 1 polymers-15-00309-t001:** Technical index of fly ash.

Fineness (45 Micron Standard Square Hole Sieve Residual)	Specific Surface Area (m^2^/kg)	Burning Loss	Water Content	SO_3_	Ca
39%	356.0	6.8%	0.6%	2.9%	0.9%

**Table 2 polymers-15-00309-t002:** Technical index of slag.

Density (g/cm^3^)	Specific Surface Area (m^2^/kg)	Liquidity Ratio	Burning Loss	Water Content	SO_3_
3.0	609.6	100%	0.8%	0.1%	0.1%

**Table 3 polymers-15-00309-t003:** Technical index of NaOH.

Mass Fraction	NaOH	Na_2_CO_3_	NaCl	Fe_2_O_3_
%	99.1	0.09	0.01	0.0022

**Table 4 polymers-15-00309-t004:** Technical index of Na_2_SiO_3_.

Na_2_O Content (%)	SiO_2_ Content (%)	Stacking Density (g/cm^3^)	Dissolution Speed (60 °C, 1%)	Whiteness (%)	Modulus
21.8	60.84	0.55	78	90.77	2.88

**Table 5 polymers-15-00309-t005:** Experimental scheme.

Test No.	Slag Content	NaOH Content	Na_2_SiO_3_ Content	Water Binder Ratio
I-1	20%	4.0%	5%	0.42
I-2	30%	4.0%	5%	
I-3	40%	4.0%	5%	
I-4	50%	4.0%	5%	
II-1	40%	2.0%	5%	
II-2	40%	2.5%	5%	
II-3	40%	3.0%	5%	
II-4	40%	3.5%	5%	
II-5	40%	4.0%	5%	
II-6	40%	4.5%	5%	
II-7	40%	5.0%	5%	
III-1	40%	4.0%	0	
III-2	40%	4.0%	1%	
III-3	40%	4.0%	2%	
III-4	40%	4.0%	3%	
III-5	40%	4.0%	4%	
III-6	40%	4.0%	5%	
III-7	40%	4.0%	6%	

**Table 6 polymers-15-00309-t006:** Weights and bias coefficients of each feature.

Y	Feature 1	Feature 2	Feature 3	Feature 4	Feature 5	Feature 6	Feature 7	Feature 8	Feature 9	Bias Coefficients
1d	1.4612	−0.3113	−0.4071	−1.0634	0.7269	0.2336	0.2751	−0.5381	0.6204	7.9744
3d	0.5399	−0.1896	−0.0833	0.5121	0.2101	0.1499	−0.0512	−0.1727	−0.1555	11.6260
7d	0.9862	1.0502	−0.4319	−0.3601	1.4809	0.0088	−2.4239	0.1161	0.3029	15.1971
28d	−0.7504	2.9924	1.1737	1.7102	1.8524	0.7939	−6.2469	2.5939	−4.3043	20.5202

Note: Features 1 to 9 are, respectively, slag, NaOH, Na_2_SiO_3_, slag^2, slag*NaOH, slag* Na_2_SiO_3_, NaOH^2, NaOH* Na_2_SiO_3_, Na_2_SiO_3_^2.

**Table 7 polymers-15-00309-t007:** COD and MSE value of predicted and actual value.

Mechanical Properties	COD		MSE	
	Training Procedure	Test Procedure	Training Procedure	Test Procedure
1d	0.8238	0.8884	0.2378	0.1398
3d	0.7608	0.8905	0.5929	0.2318
7d	0.7409	0.8602	0.9503	0.4410
28d	0.8803	0.9150	1.0873	0.7263

## Data Availability

The data presented in this study are available within this article.

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
