# Peer review of "Mechanical Properties of Fly Ash-Slag Based Geopolymer for Repair of Road Subgrade Diseases"

_polymers, 2023, doi:10.3390/polym15020309_

Round 1
Reviewer 1 Report
The paper aims to present the mechanical properties and fluidity of fly ash-slag based geopolymer grouting material and to establish a model to effectively predict the mechanical properties of geopolymer grouting material. The paper is well organized and the results are interesting. This paper can be accepted after minor revisions. The detailed suggestions are listed as follows.
1. The values in Table 7 have 3, 4, and 5 decimal places. Please standardize the number of decimal places to 3 or 4 or 5.
2. English language and style are fine/minor spell check required.
3. Please check the singular and plural forms in the paper, e.g. line 201, “Research” should be modified to “Researches”.
4. The format of references needs to be unified, e.g. Ref.[23].
5. Some of the analyses in this paper have no specific values and are not intuitive enough. For example, in section 3.1.2 it is stated that "The reason why the early strength fluctuates with the change of Na2SiO3 content is that when the content of Na2SiO3 is small, ... When the Na2SiO3 content is too high, ... (lines 344-347)".
6. There are many types of grouting materials, and only "cement-based grouting materials" are mentioned in the paper. What are the advantages of this geopolymer grouting material compared with other grouting materials?
7. Two methods for preparing geopolymer are mentioned in this paper. Please explain the difference between "one-step method" and "two-step method".
Reviewer 2 Report
The topic of the manuscript is Mechanical Properties of Fly Ash-slag Based Geopolymer for Repair of Road Subgrade Diseases.
Many authors are devoted to the topic of research, and it is necessary to significantly improve the broader context of information and motivation for research.
The presented research mainly involves experimental tests.
The experimental program is well designed, but it could be more extensive.
It is possible to test samples of the microstructure up to structural elements.
In the Introduction section, it is necessary to provide all significant information. There is extensive research in the field of geopolymers in the world. Give the information in the context of your research and state the motivation.
Interesting articles include:
Mohammed, A.A.; Ahmed, H.U.; Mosavi, A. Survey of Mechanical Properties of Geopolymer Concrete: A Comprehensive Review and Data Analysis. Materials 2021, 14, 4690.
or on selected aspects:
Bilek, V.; Sucharda, O.; Bujdos, D. Frost Resistance of Alkali-Activated Concrete—An Important Pillar of Their Sustainability. Sustainability 2021, 13, 473.
Figure 1. Improve the description and discussion of the figure in the manuscript
Figure 2. the text in the image must be enlarged
Figure 3. Enlarge the image and enlarge the text in the image
Figure 4. Enlarge the image and enlarge the text in the image
Check the Manuscript Template to be in MDPI format - Headings, Tables, ....
Figure 10. Enlarge the image and enlarge the text in the image
Figure 12. Improve the quality and visual image - processing quality.
Figure 15. Improve the quality and visual image - processing quality.
Figure 16. Enlarge the image and enlarge the text in the image
Figure 17. Enlarge the image and enlarge the text in the image
In the text of the manuscript, better comment on SEM and discuss Figures 23, 24,25.
Experiments and measurements are well documented.
The results discussion section is missing and needs to be added.
The results of the experimental program must be discussed in detail.
Improve the conclusion part.
After editing, the research and the article will be interesting for readers. In the experimental program, the authors did a good job. However, the manuscript needs to be improved.
Reviewer 3 Report
The manuscript entitled: “Mechanical Properties of Fly Ash-slag Based Geopolymer for Repair of Road Subgrade Diseases” is in line with the Polymer journal. The article based on original research. Overall, the article is well written and the investigation is on good level, but it requires following changes before publication.
· Abstract – please verify this sentence: “. Also there is no curing requirement” (line 15), because it is not true. Authors mentioned about the curing process in the same article in line 195.
· Abstract – please add measurable results.
· Introduction - This part should be better structured and shortened. The part about one- or two-step geopolymers is not justified. There is not connected with the topic of this article and should be shortened or removed. Instead, some factors, such as FA composition, that influence geopolymer properties should be briefly discussed, https://doi.org/10.3390/polym14101954
· Introduction – a lot of claims included in this part require to be better confirmed (source / reference required), including: lines 35-37, 59-62, 91-94, and 131-133.
· Introduction – “…slag resource utilization rate of 100% abroad…” please specify the mentioned country.
· Introduction (last paragraph): information on literature gaps should be clarified; the novelty aspects of the provided research should be provided.
Round 2
Reviewer 2 Report
The manuscript has been revised and improved.